# Relationships between Body Condition Score (BCS), FAMACHA©-Score and Haematological Parameters in Alpacas *(Vicugna pacos),* and Llamas *(Lama glama)* Presented at the Veterinary Clinic

**DOI:** 10.3390/ani11092517

**Published:** 2021-08-27

**Authors:** Matthias Gerhard Wagener, Saskia Neubert, Teresa Maria Punsmann, Steffen B. Wiegand, Martin Ganter

**Affiliations:** 1Clinic for Swine, Small Ruminants, Forensic Medicine and Ambulatory Service, University of Veterinary Medicine Hannover, Foundation, Bischofsholer Damm 15, 30173 Hannover, Germany; Saskia.neubert@tiho-hannover.de (S.N.); opunsma@gmail.com (T.M.P.); Martin.ganter@tiho-hannover.de (M.G.); 2Department of Anesthesiology, University Hospital, LMU Munich, Marchioninistraße 15, 81377 Munich, Germany; Steffen.wiegand@med.uni-muenchen.de

**Keywords:** South American camelids, anaemia, nutritional status, emaciation, clinical scores, haematology

## Abstract

**Simple Summary:**

Alpacas and llamas are increasingly presented at the veterinary clinic in Germany. Owners often notice too late when their animal is emaciated or anaemic. Emaciation can be detected by checking the so-called body condition score (BCS). An indication of anaemia can be provided directly by the FAMACHA©-score (FS), which has been adopted from small ruminants. There is still little information on both scores for alpacas and llamas, so a retrospective evaluation of data from the veterinary clinic was carried out. More than half of the animals admitted to the clinic were too lean and more than one in ten alpacas or one in five llamas showed clinical signs of anaemia. Both scores were compared with the findings from the animals’ blood counts, which showed that poor nutritional status was associated with anaemia and shifts in inflammatory cells. Regular monitoring of BCS and FS is therefore important in alpacas and llamas to detect emaciation and anaemia in time.

**Abstract:**

South American camelids (SAC) are being more and more presented at the veterinary Clinics in Germany. A bad nutritional condition, which can be easily categorized using a body condition score (BCS) of the animals, is often not noticed by the owners. Further anaemia is also often only detected in an advanced stage in SAC. Clinical detection of anaemia can be performed by assessing the FAMACHA©-score (FS), that is adapted from small ruminants. So far, there is only little information available about BCS and FS in SAC. In this study, both clinical scores were assessed in alpacas and llamas presented at the veterinary clinic and compared with the haematological parameters from the animals. The data were extracted retrospectively from the animals’ medical records and compared statistically. More than half of the alpacas (60%) and llamas (70%) had a BCS < 3, while 12% of the alpacas and 21% of the llamas had a FS > 2. A decreased BCS was associated with a decrease in haematocrit, haemoglobin, lymphocytes, and eosinophils, as well as an increase in FS and neutrophils. BCS and FS should be assessed regularly in SAC to detect emaciation and anaemia in time.

## 1. Introduction

Husbandry of alpacas and llamas (SAC: South American camelids) is becoming more and more popular in Europe [1,2,3]. However, it remains challenging for the owners to detect health problems. Delayed treatment, especially in connection with malnutrition, is common, since SAC are stoic and have a thick fibre coat, [4]. In some cases, emaciated SAC additionally suffer from life-threatening anaemia that requires immediate blood transfusion [5].

Body condition scoring (BCS) is a common clinical tool to detect emaciation in ruminants and SACs [4,6,7,8,9]. For clinical detection of anaemia, the close observation of the colour of the mucous membranes, especially the conjunctives, can be used [10,11].

Body condition scoring was originally developed for sheep and dairy cattle [6,7,8] but has been adapted for SAC. Different body scoring-systems have been evaluated for alpacas and llamas, which include visual and palpatory examination of different body sites such as the lumbar spine, withers, shoulders, ribs, front and rear legs, or the pelvis [4,9,12,13,14,15]. A scale from 1 to 5 (1 = emaciated; 2 = thin; 3 = optimal; 4 = overweight; 5 = obese) is commonly used for determination of the nutritional status [4,15,16]. An optimal BCS (3) can be characterized by a straight line between the dorsal spinous and transverse processes of the lumbar spine [15,16]. A more concave line would be interpreted as a lower BCS, a more convex line as a higher BCS [15,16]. Causes for a bad nutritional status in SAC include management problems such as a restricted animal/feeding place ratio, insufficient nutrient supply, infestation with endoparasites, tooth problems, or any other chronical disease [4,17,18,19]. To date, there are several descriptions about the assessment of BCS in SAC, and there is also some information available about the relationship between BCS and health in llamas and alpacas [11,19,20]. However, so far, there is a lack of data concerning the BCS of SAC presented in the veterinary clinic.

In addition to malnutrition, anaemia is frequently observed in SACs [5,11], which is often caused by haemonchosis (*Haemonchus contortus*) [11,21]. Other reasons for anaemia in llamas and alpacas include haemotroph mycoplasms (*Candidatus mycoplasma haemolamae*) [22,23], gastric ulcers [24], or undersupply with trace elements that can lead to copper-, cobalt-, or iron deficiency [25,26,27]. If symptomatic, anaemia can lead to a life-threatening state, requiring immediate treatment [5,28]. Early detection of anaemia is crucial, to monitor the health of a herd and as a marker for anthelmintic treatment response.

The FAMACHA©-score (FS) (“FAMACHA” is an acronym for Dr. Francois “FAffa” MAlan, who created a CHArt with pictures of the conjunctives of sheep with five different red shades) has been established for targeted selective treatment of sheep with haemonchosis [10], which is a major cause of anaemia in sheep. The FS is an easy to use tool, for the clinical detection of anaemia [29]. The different red shades of the chart should be compared with the conjunctives of the animal to be examined and are meant to reflect different packed cell volumes (PCV’s). The physiological red colour is expressed as a FS of 1 and the palest, almost white colour as a FS of 5 [10]. FS 1 and 2 are considered as “optimal” and “acceptable”, respectively, while FS 3 as “borderline” and FS 4 and 5 as “dangerous” or “fatal” [10]. The FS has been evaluated for small ruminants in many studies worldwide, so far [30,31,32,33,34,35,36,37]. Since the FS targets anaemia and not the potential underlying condition leading to it, scoring of the conjunctives is also suggested as a tool in the diagnosis of other infections that cause anaemia such as fasciolosis [38] or bovine trypanosomosis [39]. Furthermore, there are many other causes for pale conjunctives, for example, acute bleeding.

The FS has also been used as a clinical tool for detecting anaemia in SAC [11,40,41,42,43]. Storey et al. investigated the FS in 347 alpacas and 502 llamas on different farms [11]. In 17/21 farms included in their study, *Haemonchus contortus* was the predominant nematode parasite [21]. They found significant associations between the PCV and FS and between the BCS and faecal egg count (FEC), and concluded that the FS is a useful tool to detect anaemia in SAC suffering from haemonchosis [11].

In ruminants, BCS and FS are well established as common clinical examination methods. They can be collected easily and without complex technical equipment directly on site on the animal. However, there are no data yet on the relationships between the BCS and FS and the haematological parameters for SAC presented to the veterinary clinic. In this study, we evaluated those conditions in alpacas and llamas presented to the clinic, to determine the extent to which clinical findings are associated with laboratory diagnostic parameters. In contrast to previously available data collected from animals in herds in the field [11], this study involves data from sick animals. In addition, further haematological parameters such as the individual leukocyte fractions will be considered as a supplement to the previously known data. The resulting findings could provide important information for veterinarians in the field who do not have the possibility of an immediate haematological examination.

## 2. Material and Methods

### 2.1. Data Collection

Relevant data of alpacas and llamas were extracted from the medical files from the archives of the Clinic for Swine, Small Ruminants and Forensic Medicine of the University of Veterinary Medicine Hannover, Germany to an Excel sheet (Microsoft^®®^ Excel^®®^ for Office 365). Patient files were archived as paper files until August 2016 and as digital files in the patient management program “easyVET” (VetZ. EasyVET. Available online at: https://www.vetz.vet/de-de/easyVET, accessed on 29 July 2021) from September 2016 onwards. In this retrospective study, data were assessed between July 2014 and March 2021, since the recording of BCS and FS had been routinely performed during the clinical examination of SAC from July 2014. All the data used in this study were collected during veterinary diagnostic procedures after the owners had given written consent.

### 2.2. Inclusion Criteria

Only animals with information on species, gender, age, BCS, PCV, haemoglobin (Hb), total leucocytes (white blood count: WBC), and with blood smear results for cell differentiation were included in the analysis. The absolute bodyweight in kg and FS were not recorded for each animal. Therefore, they were only evaluated in a subset of animals.

### 2.3. Collected Parameters from the Animals

#### 2.3.1. Basic Data about the Animal

Basic data included the animal’s clinic-ID, species (alpaca or llama), gender (female or male), day of examination, animal’s birthday, animal’s age (in days, calculated by subtracting the animal’s birthday from the day of examination), and, if available, the animal’s bodyweight in kg.

#### 2.3.2. Clinical Scores

Both evaluated clinical scores (BCS and FS) were assessed during the first clinical examination immediately after the animal had been presented to the clinic.

##### BCS

The BCS was noted as a score from 1 (emaciated) to 5 (obese) with 0.5 steps in between. BCS was assessed by experienced examiners by palpation of the lumbar spine according to the method described recently [15]. An optimal BCS was defined as 3.

##### FS

The FS was noted as a score for the colour of the mucous membranes of the conjunctives from 1 (physiological red colour) to 5 (pale, almost white). The red shades in between were expressed as scores 2, 3, and 4. FS was assessed by experienced examiners, by presenting the lower palpebral conjunctives. The official FAMACHA©-chard was used for learning the technique, but in most of the cases in the routine the FS was assessed as a subjective impression of the conjunctival colour, without using the colour standard of the card as a direct comparison. A FS > 2 was defined as a hint of anaemia [10]. In animals with reddened conjunctiva due to conjunctivitis, for example, no FS could be assessed.

#### 2.3.3. Haematological Parameters

Routine EDTA-blood samples (EDTA Monovette 9 mL K3E, Sarstedt AG & Co. KG, Nümbrecht, Germany) were taken from the jugular vein from each animal during the general clinical examination [44]. Blood samples were either processed directly or stored at 4 °C when the animals were presented at night or during the weekend.

##### PCV [L/L]

The packed cell volume (PCV), haematocrit was evaluated by centrifugation of EDTA-blood in a microhematocrit tube for 10 min at 10,000× *g*. The PCV was determined as the ratio of the volume of red blood cells divided by the total blood volume.

##### Haemoglobin (Hb) [g/L]

Hb was determined photometrically using a cyan solution [45]. A total of 10 µL EDTA blood was added to the 2.5 mL cyan solution (containing 18 mmol/L sodium hydrogen carbonate, 0.768 mmol/L potassium cyanide, 0.608 mmol/L potassium ferricyanide) in a cuvette. The solution was incubated for 5 min and then measured with a photometer (546 nm). Hb in g/L was determined by multiplying the determined extinction with the factor 368.

##### Total Leucocytes/White Blood Count (WBC) [G/L]

WBC was determined microscopically in a Neubauer counting chamber after 5 min lysis of 100 µL EDTA blood in 900 µL 3% acetic acid solution [45]. If normoblasts were observed in the subsequent differentiation of the blood smear, the total leukocyte count was mathematically corrected according to the following formula: WBC [G/L] = number of nuclei counted in the Neubauer chamber [G/L] × 100/(100 + Number of normoblasts per 100 leukocytes).

##### Lymphocytes, Segmented Neutrophils, Band Neutrophils, Eosinophils, Basophils, Monocytes, Normoblasts, all [%]

Differentiation of leucocytes was performed microscopically in a blood smear stained according to Pappenheim. In each blood smear, 200 leukocytes were differentiated according to their morphological features and assigned to lymphocytes, segmented neutrophils, band neutrophils, eosinophils, basophils, and monocytes. Normoblasts (nucleated red blood cells) that occurred during differentiation were recorded in addition to the 200 leucocytes.

### 2.4. Statistical Analysis

Statistical tests were performed using SAS Enterprise Guide 7.1. Descriptive statistics were expressed as median (med), mean, standard deviation (SD), minimum (min), maximum (max), lower quartile (LQ), and upper quartile (UQ) of the investigated parameters in each group. Normal distribution was tested with the Shapiro-Wilk test.

The Wilcoxon-two-sample-test was used to test for statistical differences between alpacas and llamas, as well as differences in gender (male or female) or age (crias [<1 year] or adults [>1 year]) for both species separately. Furthermore, anaemic and non-anaemic animals were compared separately in both species. The reference values of Hengrave-Burri et al. [46] were used to categorize anaemic and non-anaemic animals. Animals whose PCV were below the corresponding lower reference value (alpacas: All crias: 0.29 L/L, adult males: 0.29 L/L, adult females: 0.26 L/L; llamas: All crias: 0.28 L/L, adult males: 0.29 L/L; adult females: 0.27 L/L) were classified as anaemic, while all other animals were classified as non-anaemic.

The Kruskal-Wallis test was used to determine differences for different BCS or FS. Spearman’s rank correlation coefficient was used for calculating correlations between the investigated parameters and BCS or FS.

A p-value less than 0.05 was considered significant (* = *p* < 0.05; ** = *p* < 0.01; *** = *p* < 0.001). Significant correlations were interpreted as follows: R = 0.2–0.3: Very weak correlation; R = 0.3–0.5: Weak correlation; R = 0.5–0.8: Moderate correlation; R > 0.8: Strong correlation. Since most of the investigated groups had a value of *p* < 0.05 in the Shapiro-Wilk test and therefore, failed normal distribution, nonparametric tests were performed. In the descriptive data (Appendix A), both the median and mean were given, since the differences in the mean were more obvious.

## 3. Results

### 3.1. Population

The total number of all animals (N = 300) was further divided into the species (al-paca (A) or llama (L)), gender (male or female), and the age (crias: < 1 year; adults: > 1 year). A total of 10 different groups were considered:All alpacas (n = 259)All llamas (n = 41)Alpacas, male, cria (n = 26)Alpacas, male, adult (n = 87)Alpacas, female, cria (n = 40)Alpacas, female, adult (n = 106)Llamas, male, cria (n = 2)Llamas, male, adult (n = 21)Llamas, female, cria (n = 2)Llamas, female, adult (n = 16)

### 3.2. Clinical Parameters

#### 3.2.1. BCS

About 60% of the alpacas (n = 154/259) and 70% of the llamas (n = 29/41) had a BCS of lower than three. Alpaca crias showed a lower BCS than adult alpacas, with no differences between species or gender (Table 1 and Appendix A). Alpacas or llamas with anaemia had a lower BCS than alpacas and llamas without anaemia (Table 1 and Appendix A). When comparing animals with different BCS, there were significant differences in bodyweight, FS, PCV, Hb, and eosinophils in both species (Appendix A). In llamas, there were additionally differences in lymphocytes, segmented neutrophils, and basophils depending on the BCS of the animal (Appendix A).

BCS in alpacas correlated a weak positive correlation with bodyweight and very weak with PCV, Hb, and eosinophils (Table 2 and Appendix A). A weak negative correlation was observed for BCS and FS (Table 2 and Appendix A).

In llamas, BCS correlated a moderate positive correlation with bodyweight, PCV, Hb, lymphocytes, and eosinophils, whereas FS and segmented neutrophils revealed a negative correlation with BCS in llamas (Table 3 and Appendix A).

#### 3.2.2. FS

About 12% (n = 24/214) of the investigated alpacas and 21% (n = 8/39) of the investigated llamas had an FS > 2. When comparing species or gender separately for each species, there were no significant differences regarding FS (Table 1, Appendix A), but there was an effect of age in alpacas: Crias had a lower FS than adult alpacas (Table 1, Appendix A). In both species, animals with anaemia had a significantly higher FS than animals without anaemia (Table 1, Appendix A). Different FS were associated with differences in BCS, PCV, Hb, MCHC, lymphocytes, eosinophils, and normoblasts in alpacas and with differences in BCS, PCV, Hb, eosinophils, and normoblasts in llamas (Appendix A).

In alpacas, weak negative correlations were found between FS and BCS, PCV and Hb, and a very weak positive correlation between FS and normoblasts (Table 2 and Appendix A). In llamas, moderate negative correlations were found between FS and BCS, PCV, Hb, and eosinophils and a weak negative correlation between FS and bodyweight. FS in llamas further correlated moderate positively with normoblasts (Table 3 and Appendix A).

### 3.3. Haematological Parameters

#### 3.3.1. PCV

When compared with the reference values of Hengrave Burri (2005) [46], the majority of SAC that was presented to the clinic suffered from anaemia (55% of the alpacas, 49% of the llamas). The lowest PCV in a single alpaca or llama was 0.04 L/L, the maximal PCV was 0.43 L/L in an alpaca and 0.47 L/L in a llama (Appendix A). Differences for PCV in species, gender, or age were not statistically significant (Table 1 and Appendix A). In both alpacas and lamas, a low PCV was found to be associated with a low BCS (Figure 1) and a higher FS (Appendix A). PCV in alpacas revealed a strong positive correlation with Hb, a weak positive correlation with BCS and monocytes, as well as a weak negative correlation with FS and normoblasts (Table 2 and Appendix A). PCV in llamas also revealed a strong positive correlation with Hb, a moderate positive correlation with bodyweight, BCS, and eosinophils, as well as a moderate negative correlation with FS and normoblasts (Table 2 and Appendix A). All the animals with severe anaemia (PCV < 0.10 L/L) had a BCS ≤ 3.

#### 3.3.2. Hb

Hb in alpacas had a range of 20–198 g/L, while in llamas, the range was 15–218 g/L (Appendix A). Since PCV and Hb showed a very strong correlation, Hb showed similar associations with the other parameters as PCV (Table 2 and Table 3).

#### 3.3.3. WBC

The range of WBC was 0.3–75.0 G/L in alpacas and 2.8–40.4 G/L in llamas (Appendix A). Differences in species or gender were not detected, nor was there a difference between alpaca crias and adult alpacas or anaemic and non-anaemic animals (Table 1 and Appendix A). Influences of different BCS or FS were not evident in the Kruskal-Wallis test (Appendix A) and correlations with BCS, FS, and PCV did not yield significant results (Table 2, Table 3, Appendix A).

#### 3.3.4. Lymphocytes

Lymphocytes revealed a range of 1–90% in alpacas and 3–65% in llamas (Appendix A). Lymphocyte percentages were higher in alpacas than in llamas and higher in alpaca crias than in adult alpacas (Table 1 and Appendix A). Differences between gender or anaemic and non-anaemic animals were not observed (Table 1, Appendix A). The proportion of lymphocytes was different depending on the BCS of the llamas (Appendix A). BCS and lymphocytes correlated as moderate positive in llamas (Figure 2). However, in alpacas, there was no significant correlation between BCS and lymphocytes (Table 3 and Appendix A).

#### 3.3.5. Neutrophils

The segmented neutrophils´ ranges were 3–98% in alpacas and 19–95% in llamas (Appendix A). The proportion of segmented neutrophils in alpaca crias was lower than in adult alpacas (Table 1, Appendix A), but there were no statistical differences between species, gender, or animals with and without anaemia for segmented neutrophils (Table 1, Appendix A). For band neutrophils, there was a statistical difference between anaemic and non-anaemic animals in alpacas. However, the numerical proportion of band neutrophils was lower in anaemic alpacas than in alpacas without anaemia, whereas the opposite was the case in llamas.

In llamas, BCS was associated with segmented neutrophils (Appendix A). Segmented neutrophils in llamas with BCS of 1 revealed the highest median and segmented neutrophils in llamas with BCS 3 the lowest (Appendix A and Figure 2). The correlation between BCS and segmented neutrophils in llamas was weak negative (Table 3 and Appendix A), however, this was seen only in lamas, not in alpacas (Table 2 and Appendix A). Different segmented neutrophils were not reflected in the FS of alpacas or llamas (Appendix A).

#### 3.3.6. Eosinophils

The percentages of eosinophils were 0–30% in alpacas and 0–39% in llamas (Appendix A). There were no differences in eosinophils concerning species or gender (Appendix A), but adult alpacas had significantly higher amounts of eosinophils than alpaca crias (Table 1 and Appendix A). In addition, llamas with anaemia had a lower percentage of eosinophils than llamas without anaemia. However, this relationship was not seen in alpacas (Table 1, Appendix A). In both species, a relationship between BCS and the proportion of eosinophils existed, with both alpacas and llamas having the lowest median of eosinophils at BCS 1 (Appendix A). The correlation between BCS and eosinophils was stronger in llamas than in alpacas (Table 2, Table 3 and Appendix A). Different FS went in hand with different proportions of eosinophils. Nonetheless, this was only significant for llamas (Appendix A) and was also reflected in a moderate negative correlation between FS and eosinophils in llamas (Table 3 and Appendix A). Further eosinophils in llamas correlated moderately positively with PCV (Table 3 and Appendix A).

#### 3.3.7. Basophils

Basophils had a range of 0–4% in alpacas and 0–3% in llamas (Appendix A). There were no statistical differences concerning species, gender, and age or between anaemic and non-anaemic animals (Table 1 and Appendix A). However, BCS was associated with the proportion of basophils in llamas (Appendix A). Nevertheless, there were no significant correlations between basophils and BCS, FS or PCV in either species (Table 2, Table 3, Appendix A).

#### 3.3.8. Monocytes

The monocyte ranges were 0–21% for alpacas and 0–9% for llamas (Appendix A). There were no statistical differences between species, age, or gender (Table 1 and Appendix A), but alpacas with anaemia revealed lower proportions of monocytes in the differential count than alpacas without anaemia (Table 1, Appendix A). However, this was not the case in llamas. Different BCS or FS were not reflected in the proportion of monocytes (Appendix A) and there were no significant correlations between monocytes and BCS, FS, or PCV in alpacas or llamas (Table 2, Table 3, Appendix A).

#### 3.3.9. Normoblasts

Normoblasts were present in 33% (n = 85/259) of the alpacas and 34% (n = 14/41) of the llamas. Species, gender, and age had no effect on normoblasts (Table 1 and Appendix A). In anaemic llamas, the amount of normoblasts was higher than in non-anaemic llamas (Table 1, Appendix A). However, this was not statistically significant for alpacas. Different BCS had no impact on normoblasts, but FS was connected to normoblasts (Appendix A). This was also reflected in a positive correlation between FS and normoblasts, that was moderate in llamas but only very weak in alpacas (Table 2, Table 3 and Appendix A). Normoblasts further correlated negatively with PCV in both species (Table 2, Table 3 and Appendix A).

## 4. Discussion

We found that most of the llamas and alpacas presented to the clinic had a low BCS and about half of all alpacas and llamas were anaemic. For the clinical parameters, we showed that a low BCS was associated with lower body weight and increased FS in both species. When comparing this with the haematological parameters, a low BCS was also associated with decreased PCV, Hb, and eosinophils in both species. In llamas, a low BCS was additionally associated with a lower percentage of lymphocytes and an increased percentage of segmented neutrophils. In addition, despite the different age of the animals, a good correlation of bodyweight and BCS was found in both species.

A main finding in alpacas and llamas at the clinic is that a poor nutritional status is usually related to a low BCS and in most cases anaemia, as well. The condition of the animals is rarely perceived by the owners themselves. One cause for this is the very dense hair coat of SACs. As the owners do not palpate their animals regularly, the emaciation may remain undetected [4]. In a survey among alpaca and llama owners in Germany [1], fewer than half of all participating 255 farms (38.9%) reported emaciation in their animals from the owners´ observation. Anaemia was observed even less frequently: Only 13.3% of the farms reported anaemia in their animals from their own observation [1], which contrasts with the results of the present study. The numbers cannot be easily compared since the animals presented to the clinic are usually sick and most healthy animals from the herds are never presented to the clinic. However, Storey et al. found a much lower proportion of animals with a BCS < 3 in their study on alpaca and lama farms than in the present study. Nonetheless, these animals with lower BCS were overrepresented in the group of anaemic animals [11], which is in line with our data. Storey et al. also found a higher proportion of llamas with anaemia than alpacas. Although this disagrees with our findings, it could be explained by the fact that mainly animals with pathological conditions are presented to the clinic [11].

For detecting anaemia in SAC with a PCV ≤ 0.17 L/L with an FS ≥ 4, a sensitivity of 50% and a specificity of 94% are given by Storey et al. [11]. FS showed a significantly negative correlation with PCV, Hb, as well as BCS, especially in the llamas.

It should be noted, however, that the PCV determined per FS has a wide range. This range can be explained by the fact that the FS does not specifically assess the PCV as it can also be elevated or depressed due to local inflammatory processes in the conjunctives or circulatory centralisation [34]. In addition, there is also a lack of specific colour scales for SAC. Since the FS was routinely assessed at the clinic by several examiners, this factor of differing expert opinions must also be taken into consideration. No data are available on this for SAC, but the studies by Maia et al. regarding the assessment of different examiners on the FS in sheep and goats indicate that the FS can be collected quite accurately by different persons after adequate training [47]. For everyday clinical practice, a less precise subdivision (for example “physiological red”, “pale”, and “white”) of the FS might be sufficient to gain a first impression of a single animal.

Similar positive correlations of BCS and PCV or Hb as found in our study were also described in sheep or cattle. In pre- and post-partum cattle, Rafia et al. determined r = 0.32 and r = 0.36 for the correlation of BCS and PCV [48]. Torres-Chable et al. investigated the correlation of BCS and PCV in sheep and found r = 0.39 [49]. It is also noteworthy that in this present study, none of the animals with a BCS >3 had severe anaemia (<0.10 L/L).

The association of low BCS and anaemia remains unclear, there could be speculation about atrophy of the bone marrow or chronic inflammation. However, it must also be considered that the animals were usually transported prior to blood sampling and stress reactions can also lead to shifts in leukocyte fractions. Although no statistically relevant associations with WBC and BCS or PCV were found, the shift in the proportions of leukocytes from lymphocytes to segmented neutrophils in animals with low BCS, especially in the llamas, may provide indications for inflammation. A decrease in lymphocytes was also observed in emaciated horses [50].

Reference values play an important role when interpreting laboratory results. The reference values consulted in this study had a lower limit of 0.26–0.29 L/L for PCV, depending on species, gender, or age [46], and were thus in a similar range to reference values for alpacas and llamas from other authors [51,52,53]. In the animals in our study, no differences in PCV were found with respect to age or gender. However, other authors showed that male SAC usually have higher PCV than females [54,55]. This could be due to the fact that our study did not investigate homogeneous groups, but rather data from animals with different pathological conditions. Moreover, this could possibly be the reason why no significant relationships were found between WBC and the other parameters. However, Rafia et al. found only a very weak negative correlation (r = −0.15) for BCS and WBC in cattle [48]. When interpreting the PCV, it should also be taken into account that the blood samples were usually taken after the animals had been transported. Here, the transport stress could have led to haemoconcentration effects in individual animals.

The positive correlation between BCS and eosinophils remains unclear. In studies on the influence of BCS on haematological parameters in other species, such a relationship was not reported [48,49]. Since anaemia and poor nutritional status are often associated with endoparasites in alpacas [11,21], a negative correlation was expected. It could be speculated here that animals with a lower BCS are in a more immunologically inactive state, which is particularly reflected in the number of eosinophils. However, evidence for this assumption is lacking.

A lower percentage of monocytes in anaemic animals could be due to infection or stress. Nonetheless, since monocytes are generally present in low numbers in a blood smear, a single over- or under-recognized cell during differentiation can account for a large error. The same also applies to the basophils, which are typically found only very sporadically in the blood smear.

Normoblasts or erythroblasts are usually associated with regenerative anaemia, but there are only few exceptions where normoblasts may appear in the peripheral blood in the absence of anaemia [56]. Although there was a clear negative correlation between PCV and normoblasts, it remains questionable why normoblasts were found equally in anaemic and non-anaemic alpacas. Normoblasts can be present in large numbers in alpacas with highly regenerative anaemia, where extramedullary haematoopoiesis seems to play a role [57]. However, there are still too few data on normoblasts in alpacas in general, and more research is needed. Another haematological parameter that provides information about regeneration is the reticulocyte count [58], but this was only determined in some of the animals investigated in our study and was excluded from the evaluation due to the low number.

## 5. Limitations

The focus of this study was to compare clinical scores (BCS and FS) with haematological findings. Therefore, this does not allow a general statement concerning individual clinical pictures. It is also not possible to draw a conclusion whether a lower BCS was the cause of haematological changes, or whether the BCS was changed as a result of changes that were visible in the blood count. Furthermore, the results of faecal egg counts were not considered. Several animals presented to the clinic had already been dewormed by other veterinarians or by the owners themselves, shortly before presentation at the clinic. This made it impossible to determine the overall contributory of endoparasites to the clinical picture. The clinical diagnoses of the animals were not taken into account in the evaluation. It was not possible to define a main diagnosis for every animal, as some animals had more than one diagnoses. Common diagnoses besides emaciation or anaemia included recumbency or gastric ulcers. Data on pregnancy stage or lactation were also not available for all the animals. Therefore, these parameters were not taken into account.

## 6. Conclusions

In summary, more than half of the alpacas and llamas presented to the clinic had a BCS < 3. In addition, half of the animals suffered from anaemia. A low BCS was predominantly associated with increased FS and decreased PCV and Hb. There was also evidence that a low BCS was associated with an increase in segmented neutrophils, which may indicate that animals with lower BCS are more likely to be affected by inflammatory diseases.

It is important to educate SAC owners about the fact that a poor nutritional status is closely associated with pathological haematological findings, and to encourage them to check the nutritional status of the animals regularly, so that emaciation can be detected in time. Recording BCS and FS is also particularly useful for identifying potentially anaemic animals. The early detection of pathological conditions in alpacas and llamas can thus make an important contribution to animal welfare.

## Figures and Tables

**Figure 1 animals-11-02517-f001:**
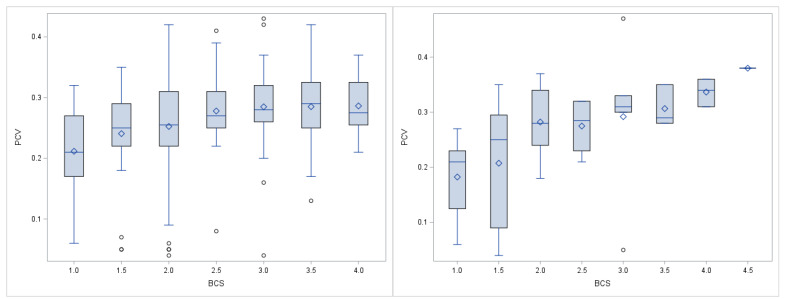
PCV in L/L in alpacas (n = 259, **left**) and llamas (n = 41, **right**) with different BCS. The boxplots display the quartile range and the respective minimum and maximum.

**Figure 2 animals-11-02517-f002:**
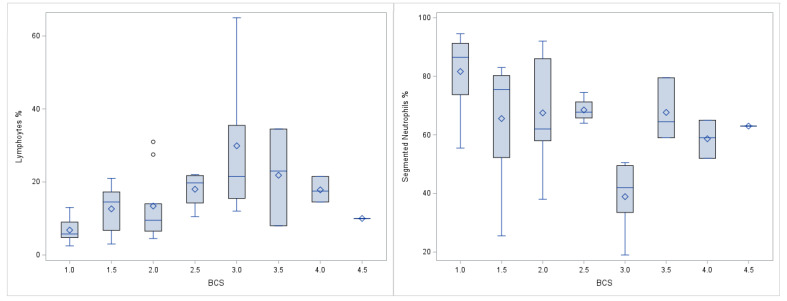
Proportions of lymphocytes (**left**) and segmented neutrophils (**right**) in llamas (n = 41) with different BCS. The boxplots display the quartile range and the respective minimum and maximum.

**Table 1 animals-11-02517-t001:** Results of the Wilcoxon test to check for differences in species (alpacas vs. llamas), gender (male vs. female), age (juvenile vs. adult), and presence of anaemia (anaemia vs. without anaemia). No data on the influence of age are available for llamas, since the juvenile llamas group contained only two animals. There were no significant differences between male and female animals for any of the parameters. * = *p* < 0.05; ** = *p* < 0.01; *** = *p* < 0.001; n.s. = not significant.

Parameter	Alpaca vs. Llama	Juvenile vs. Adult	Anaemia vs. without Anaemia	
	All	Alpacas	Alpacas	Llamas
Bodyweight (kg)	***	***	***	***
BCS	n.s.	**	***	***
FS	n.s.	*	**	***
PCV (L/L)	n.s.	n.s.	***	***
Hb (g/L)	n.s.	n.s.	***	***
WBC (G/L)	n.s.	n.s.	n.s.	n.s.
Lymphocytes (%)	**	***	n.s.	n.s.
Segmented neutrophils (%)	n.s.	**	n.s.	n.s.
Band neutrophils (%)	n.s.	n.s.	*	n.s.
Eosinophils (%)	n.s.	**	n.s.	**
Basophils (%)	n.s.	n.s.	n.s.	n.s.
Monocytes (%)	n.s.	n.s.	*	n.s.
Normoblasts (%)	n.s.	n.s.	n.s.	*

**Table 2 animals-11-02517-t002:** Results of Spearman’s correlation for BCS, FS, and PCV with the investigated clinical and haematological parameters in alpacas (n = 259). * = *p* < 0.05; *** = *p* < 0.001; n.s. = not significant.

Alpacas	BCS		FS		PCV	
	r =		r =		r =	
Bodyweight (kg)	0.41	***	0.03	n.s.	0.16	*
BCS			−0.32	***	0.29	***
FS	−0.32	***			−0.34	***
PCV (L/L)	0.29	***	−0.34	***		
Hb (g/L)	0.28	***	−0.32	***	0.94	***
WBC (G/L)	−0.08	n.s.	−0.08	n.s.	0.00	n.s.
Lymphoytes (%)	0.09	n.s.	−0.08	n.s.	0.12	*
Segmented neutrophils (%)	−0.08	n.s.	0.04	n.s.	−0.13	*
Band neutrophils (%)	−0.12	n.s.	0.13	n.s.	0.08	n.s.
Eosinophils (%)	0.28	***	−0.11	n.s.	−0.10	n.s.
Basophils (%)	0.11	n.s.	−0.12	n.s.	−0.13	*
Monocytes (%)	0.06	n.s.	0.03	n.s.	0.22	***
Normoblasts (%)	−0.09	n.s.	0.26	***	−0.20	***

**Table 3 animals-11-02517-t003:** Results of Spearman’s correlation for BCS, FS, and PCV with the investigated clinical and haematological parameters in llamas (n = 41). * = *p* < 0.05; ** = *p* < 0.01; *** = *p* < 0.001; n.s. = not significant.

Llamas	BCS		FS		PCV	
	r =		r =		r =	
Bodyweight (kg)	0.66	***	−0.36	*	0.56	***
BCS			−0.55	***	0.59	***
FS	−0.55	***			−0.73	***
PCV (L/L)	0.59	***	−0.73	***		
Hb (g/L)	0.60	***	−0.72	***	0.96	***
WBC (G/L)	−0.25	n.s.	−0.06	n.s.	0.18	n.s.
Lymphoytes (%)	0.54	***	−0.17	n.s.	0.07	n.s.
Segmented neutrophils (%)	−0.46	**	0.25	n.s.	−0.16	n.s.
Band neutrophils (%)	−0.13	n.s.	0.22	n.s.	−0.15	n.s.
Eosinophils (%)	0.61	***	−0.54	***	0.50	***
Basophils (%)	0.25	n.s.	−0.04	n.s.	0.09	n.s.
Monocytes (%)	−0.01	n.s.	−0.05	n.s.	0.05	n.s.
Normoblasts (%)	−0.30	n.s.	0.54	***	−0.53	***

## Data Availability

The datasets supporting the conclusions of this article are included within the article and its additional file. The raw data used in this study are not publicly available since they are veterinary patient records subject to confidentiality. The raw data are located in the patient archive of the Clinic for Swine, Small Ruminants, Forensic Medicine and Ambulatory Service of the University of Veterinary Medicine Hannover Foundation and were analysed with the permission of the clinic management and the Animal Welfare Officer of the University of Veterinary Medicine Foundation.

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
