# Peer review of "Relationships between Body Condition Score (BCS), FAMACHA©-Score and Haematological Parameters in Alpacas (Vicugna pacos), and Llamas (Lama glama) Presented at the Veterinary Clinic"

_animals, 2021, doi:10.3390/ani11092517_

Round 1
Reviewer 1 Report
Brief summary: The study describes associations between body condition score, FAMACHA score, and various hematological parameters in llamas and alpacas presented in a University clinic.
General comments: The clinical questions and their transposition into statistical analyses are not clear. Numerous results are presented, with some associations being evaluated twice and yielding different results. The results are poorly synthesized, making interpretation difficult. The generalizability and the relevance of the findings are unclear.
Specific comments:
Introduction
Line 86-87: What is anemia due to cachexia?
Objective
Lines 98-100: The objective could be better explained or justified, or hypotheses stated. If a relationship between FS and PCV was already established in the field, why would the population presented to the clinic differ, and/or what are additional hematological parameters to be investigated and why? What would be the clinical implications of such relationships? Furthermore, the usefulness of FS as a proxy for PCV on farms is clear, as laboratory analyses are not readily available. However I would not imagine that in the clinic, a clinical parameter would justify to forego conducting laboratory analyses.
Methods
Lines 120-121: Was gestational or lactation stage available? Motive for clinic admission, and final diagnosis category?
Line 164: I would suggest avoiding the abbreviation PMN for neutrophils, since eosinophils, etc. also are PMN.
Line 170: Why were % of differentiated WBC used rather than their absolute numbers?
Explanation and justification of statistical analyses could be improved.
Line 175: in each group = each species?
Lines 177-178: Categorization of age and anemia should be described.
Results in general could be further synthesized.
Line 190: Please start with a description of the study population (species, age, gender).
Table 1: Non-significant results (e.g. gender) could rather be mentioned in the text. It would be useful to see the actual values in the groups (e.g. anemia) as well as the reference values used.
Line 173, 299: I would avoid stating an "impact" or "influence" (direction in the association) between findings.
Discussion
Line 330: If the problem is rarely perceived by the owners, then what are the actual reasons of presentation?
Lines 364-365: This sentence is not clear. The list of possible conditions causing both low BCS and PCV is extensive. Similarly, I would be careful interpreting %PMN in possibly stressed animals that are sick, transported, and presented to the clinic.
Lines 397-399: Sentences are not clear. Do you mean that in cases with regenerative anemia, the strength of the response is correlated with the severity of anemia, whereas it would not be in cases of non-regenerative anemia (such as chronic inflammation discussed)?
Reviewer 2 Report
The paper is devoted to the important clinical health problems of South American camelids. While this is not a strictly scientific work, it provides very important clinical information that is also applicable to science. The data used in the work are numerous, which makes the research and conclusions drawn very reliable. The work is written very well.
I only have two minor comments:
1st line 48 - "Body Condition scoring" should be changed to "Body condition scoring"
2. line 67 - haemonchosis [11, 21] should be supplemented with "haemonchosisi (Haemonchus contortus) [11,21]"
